# Boosting the Uniqueness of Neural Networks Fingerprints with Informative Triggers

**Zhuomeng Zhang**
Shanghai Jiao Tong University
zzmsmm@sjtu.edu.cn

**Fangqi Li**
Shanghai Jiao Tong University
solour_lfq@sjtu.edu.cn

**Hanyi Wang**
Shanghai Jiao Tong University
why_820@sjtu.edu.cn

**Shi-Lin Wang**[*]
Shanghai Jiao Tong University
wsl@sjtu.edu.cn

## Abstract

One prerequisite for secure and reliable artificial intelligence services is tracing the copyright of backend deep neural networks. In the black-box scenario, the copyright of deep neural networks can be traced by their fingerprints, i.e., their outputs on a series of fingerprinting triggers. The performance of deep neural network fingerprints is usually evaluated in robustness, leaving the accuracy of copyright tracing among a large number of models with a limited number of triggers intractable. This fact challenges the application of deep neural network fingerprints as the cost of queries is becoming a bottleneck. This paper studies the performance of deep neural network fingerprints from an information theoretical perspective. With this new perspective, we demonstrate that copyright tracing can be more accurate and efficient by using triggers with the largest marginal mutual information. Extensive experiments demonstrate that our method can be seamlessly incorporated into any existing fingerprinting scheme to facilitate the copyright tracing of deep neural networks.

## 1 Introduction

Recent progress in deep neural network (DNN) models is raising privacy and ethics concerns since they might facilitate the propagation of fake information with negative social impacts [1]. To ensure that AI serves people properly, it is necessary to trace the copyright of DNN models and attribute the misuse of models to specific users. The majority of existing studies concentrate on copyright tracing in the black-box setting where the copyright verifier interferes with the suspicious DNN model as a black box. Two mainstream methods are DNN watermarking [2–8] and DNN fingerprinting [9–16].

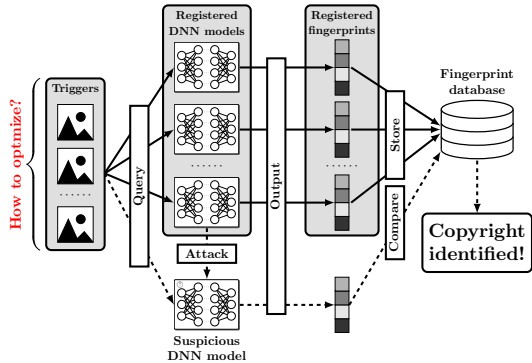

Figure 1: The framework of a fingerprint-based DNN copyright tracing system.

DNN watermarking schemes use a series of inputs as watermark triggers and tune DNN models so that their outputs on triggers are differentiated as pre-defined. Therefore, they appeal only to copyright

---

[*]Corresponding author

39th Conference on Neural Information Processing Systems (NeurIPS 2025).

verifiers who can tune the DNN models to be protected or owners who distribute a model to multiple clients. Watermarking schemes inevitably introduce deline in the performance of watermarked DNN models, which is unacceptable in industrial fields.

In contrast, DNN fingerprinting schemes produce a series of fingerprinting triggers. The outputs of a suspicious DNN model on the triggers constitute its fingerprint, with which its identity is recognized. An illustration is given in Fig. 1.

The reliability of DNN fingerprinting schemes has usually been interpreted as their robustness, i.e., the fingerprint of a DNN model should remain invariable under adversarial modifications. Despite established results on robustness, the uniqueness of DNN fingerprinting is still under-explored. This aspect is crucial for industrial copyright tracing especially when the number of model to be traced is large while the expense of retrieving a fingerprint grows in the number of triggers and could become a bottleneck. It is hard to evaluate how every trigger contributes differently to copyright tracing, and under which circumstances does each DNN model have a unique fingerprint. Consequently, it is difficult to optimize the collection of triggers with a fixed cardinality. To address the above challenges, this paper proposes a general method to improve the efficiency of fingerprint-based DNN copyright tracing. The contributions are concluded as follows:

- We adopt an information theoretical perspective to measure the contribution to copyright tracing of each fingerprinting trigger by its conditional mutual information.

- We boost the copyright tracing performance by greedily optimizing the collection of triggers and validate this method through extensive experiments.

- We derive the first necessary condition for the number of fingerprinting triggers to ensure copyright tracing.

## 2 Preliminaries

### 2.1 DNN Fingerprint

Notations used in this paper are listed in Table 1. We focus on classifiers that map the input space $\mathcal{X}$ to $\mathcal{Y} = \{1, 2, \cdots, C\}$. Altogether $P$ classifiers $\mathbf{F} = \{f_p\}_{p=1}^{P}$ require copyright tracing. A fingerprinting scheme draws $N$ independent triggers $\mathbf{T} = \{\mathbf{t}_n\}_{n=1}^{N}$ from a distribution $\mathcal{T}$. Choices of $\mathcal{T}$ include random noises [9], outliers [3, 10], normal samples that are close to the centers of each class [15, 16], adversarial samples that are close to the decision boundaries [11, 13], etc.

The fingerprint of the $p$-th model is its outputs on triggers: $(f_p(\mathbf{t}_1), \cdots, f_p(\mathbf{t}_N))$. The fingerprints of all models are computed and registered

Table 1: Frequently used notations in this paper.

| Symbol | Meaning |
|---|---|
| $\mathcal{X}$ | Input space of DNN models. |
| $C$ | Number of classes. |
| $\mathcal{Y}$ | Output space of DNN models. |
| $P$ | Number of registered DNN models. |
| $\mathbf{F}$ | Registered DNN models, $\mathbf{F} = \{f_p\}_{p=1}^{P}$. |
| $N$ | Number of fingerprinting triggers. |
| $\hat{N}$ | Number of greedily selected triggers. |
| $\mathcal{T}$ | Distribution of triggers. |
| $\mathbf{T}$ | Fingerprinting triggers, $\mathbf{T} = \{\mathbf{t}_n\}_{n=1}^{N}$. |
| $\epsilon$ | Verifier's tolerance on the adversary's attack. |
| $\mathbf{A}_\epsilon$ | Verifier's threat model. |
| $\mathcal{F}$ | A randomly selected model from $\mathbf{F}$. |
| $\mathcal{A}$ | A randomly selected attack from $\mathbf{A}_\epsilon$. |
| $\phi_n$ | The randomly selected model's prediction on $\mathbf{t}_n$. |
| $u$ | Uniqueness rate |
| $I_\epsilon \left( \mathbf{t}_n \vert \mathbf{t}_{1:(n-1)} \right)$ | Conditional mutual information of $\mathbf{t}_n$. |

in a database. Upon locating a suspicious model, the verifier computes its fingerprint and compares it with records in the database. If there exists a registered model whose fingerprint is close to the suspicious model's, then a copyright issue is reported. Otherwise, the suspicious model is registered as a new model. The distance between two fingerprints is measured by the number of triggers where two models return different outputs.

### 2.2 Robustness & Uniqueness

The performance of a DNN fingerprinting scheme is inclusively reflected in its *robustness* and *uniqueness*.

The robustness of a DNN fingerprinting scheme is characterized by the difference between registered models' fingerprints before and after adversarial modifications. Formally, the verifier assumes that the adversary has sacrificed the victim model's performance for a probability at most $\epsilon$ (it is impossible to

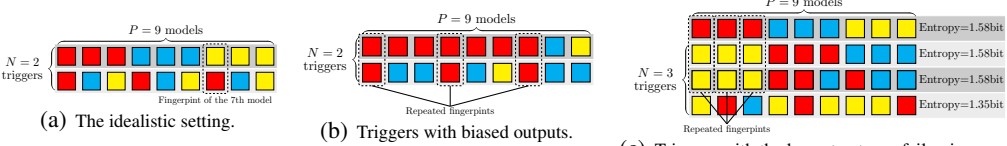

(a) The idealistic setting.  (b) Triggers with biased outputs.  (c) Triggers with the largest entropy fail uniqueness.

Figure 2: An illustration of the difficulties in deciding the optimal fingerprinting triggers $P = 9$ and $C = 3$. Each column denotes a model. Each row denotes a fingerprinting trigger. Each color denotes a classification label. (a) The idealistic case where $\log_C P = 2$ triggers produce unique fingerprints. (b) When the predictions on triggers are biased, two triggers are insufficient. (c) Choosing the first three triggers with the largest independent entropy fails the uniqueness property. The correct choice is the last three triggers.

trace the copyright of models whose functionality has been arbitrarily modified). The threat model is:

$$\mathbf{A}_\epsilon = \left\{ A : \forall f \in \mathbf{F}, \ \Pr_{x \leftarrow \mathcal{X}}(f(x) \neq f^A(x)) \leq \epsilon \right\}. \tag{1}$$

where $f^A$ denotes a model $f$ undertaken an attack $A$. In general, the adversary's attack always has a measurable influence on the model's performance so $\mathbf{A}_0 = \emptyset$. The verifier should assume a non-trivial adversary and focus on the maximal damage that the adversary might cause [17], so the robustness of a fingerprinting trigger distribution $\mathcal{T}$ is quantified by:

$$\delta_\mathcal{T}(\epsilon) = \max_{f \in \mathbf{F}, A \in \mathbf{A}_\epsilon} \left\{ \Pr_{\mathbf{t} \leftarrow \mathcal{T}} \left( f(\mathbf{t}) \neq f^A(\mathbf{t}) \right) \right\}. \tag{2}$$

A fingerprinting trigger distribution with a large $\delta_\mathcal{T}(\epsilon)$ yields weak fingerprints, since the adversary can obfuscate the fingerprints with little decline in functionality.

Empirically, the robustness of a fingerprinting trigger distribution is estimated with a finite collection of adversarial modifications including fine-tuning [18, 19], neuron pruning [20], distillation [21], etc. Although it is hard to directly foster the robustness by modifying the trigger distribution, some distributions are reported to be more robust under certain attacks [12, 15, 16].

On the other hand, the fingerprint of each model should be unique so the false positive rate is negligible. The low transferability of Characteristic Examples [22] primarily addresses the fingerprint of individual models, whereas our study focuses on the uniqueness of fingerprints for distinguishing between different models in large-scale deployment scenarios. This property is reflected in the percentage of models whose fingerprints remain unique under any attack. Let $\mathbf{F}_\epsilon(\mathbf{T}) = \{ f \in \mathbf{F} : \exists A \in \mathbf{A}_\epsilon, \exists f' \in \mathbf{F} \setminus \{f\}, \forall \mathbf{t} \in \mathbf{T}, f(\mathbf{t}) = f'(\mathbf{t}) \}$, the uniqueness rate $u$ can be defined as:

$$u = 1 - |\mathbf{F}_\epsilon(\mathbf{T})|/|\mathbf{F}|. \tag{3}$$

Unfortunately, this metric $u$ can hardly be optimized as a function in $\mathbf{T}$. In existing literature, it is generally assumed that the models' outputs on a fingerprinting trigger are randomly distributed as shown in Fig. 2(a), so $\log_C P$ triggers are sufficient and the probability that two arbitrary models have the same fingerprint declines exponentially in $N$ [9]. So any fingerprinting scheme is expected to have $u \to 1$ when $N$ is large.

## 2.3 Challenges

The assumptions behind the uniqueness of DNN fingerprinting schemes are challenged by the following facts.

**(I): The number of triggers is not arbitrarily large.** The number of triggers determines the cost of copyright tracing and could become a bottleneck in large-scale or expensive service or when querying the API of suspicious DNN models is expensive. It is necessary to evaluate the performance of DNN fingerprints when the number of triggers is limited.

**(II): The value of each trigger has been overestimated.** DNN models' outputs on a trigger might not be uniformly distributed, as shown in Fig. 2(b). Moreover, the triggers are not independent of each other. Using triggers with independently the largest entropy might turn out to be misleading as

shown in Fig. 2(c). The value of each trigger is determined by the conditional entropy it contains w.r.t. queried triggers.

**(III): The influence of adversarial modifications is unclear.** When the adversary modifies the victim model's fingerprint, the information value of each trigger might change. The optimal number of triggers varies with the assumptions of the adversary. So far, the relationship between uniqueness and robustness has not been established.

As a result, the volume of information that a limited number of triggers can provide in the adversarial environment is an intractable bottleneck of the copyright tracing system, leaving the uniqueness property as a risk.

## 3 Method

### 3.1 Information in DNN Copyright Tracing

We consider the copyright tracing of DNN models as a communication channel. From the verifier's view, the information source is the identity of the suspicious model, which is represented by a random variable $\mathcal{F}$ whose domain is $\mathbf{F}$. Without loss of generality, we assume that the suspicious model $\mathcal{F}$ is equally likely to be any registered model, so $\mathcal{F}$ contains $\log_2 P$ bits of information.

The adversarial modifications, represented by a random variable $\mathcal{A}$, are noises in this channel. Without prior knowledge, the attack is randomly chosen from $\mathbf{A}_\epsilon$, where $\epsilon$ reflects the verifier's tolerance. We further assume that the attack is independent of the triggers and the victim model.

The output from the suspicious model $\mathcal{F}$ on the $n$-th trigger, denoted by $\phi_n$, is another random variable, so is the fingerprint of the suspicious model $\Phi = (\phi_1, \phi_2, \cdots, \phi_N)$. We are interested in how much information the fingerprint reveals about the suspicious model's identity, which is inclusively quantified by the mutual information $I(\Phi; \mathcal{F})$. In fact, the volume of information to secure a uniqueness rate $u$ is at least $-u \log_2 u - (1-u) \log_2 (1-u) + \log_2 uP$, so an upper bound of $u$ is

---

**Algorithm 1** Computing $I_0\left(\mathbf{t}_n | \mathbf{t}_{1:(n-1)}\right)$.

---

**Input:** Registered models $\mathbf{F}$, triggers $\mathbf{t}_1, \mathbf{t}_2, \cdots, \mathbf{t}_n$
**Output:** $I_0\left(\mathbf{t}_n | \mathbf{t}_{1:(n-1)}\right)$
1: $\mathcal{M} = \emptyset, h = 0$
2: **for** $i = 1$ to $P$ **do**
3:     flag = False
4:     **for** M in $\mathcal{M}$ **do**
5:         **if** $\exists f \in \mathbf{F} \backslash \{f_i\}, \forall j = 1, \cdots, n-1, f(\mathbf{t}_j) = f_i(\mathbf{t}_j)$ **then**
6:             $\mathbf{M} = \mathbf{M} \cup \{f_i\}$; flag=True; break
7:         **end if**
8:     **end for**
9:     **if** flag=False **then**
10:         $\mathcal{M} = \mathcal{M} \cup \{f_i\}$
11:     **end if**
12: **end for**
13: **for** M in $\mathcal{M}$ **do**
14:     **for** $c = 1$ to $C$ **do**
15:         $u_c = 0$
16:         **for** $f$ in M **do**
17:             **if** $f(\mathbf{t}_n) = c$ **then**
18:                 $u_c = u_c + 1$
19:             **end if**
20:         **end for**
21:         $u_c = u_c/|\mathbf{M}|$; $h = h - \frac{|\mathbf{M}|}{P} \times u_c \log_2 u_c$
22:     **end for**
23: **end for**
24: **Return** $h$

---

$$u = \frac{2^{\log_2 uP}}{P} \leq \frac{2^{-u \log_2 u - (1-u) \log_2(1-u) + \log_2 uP}}{P} \leq \frac{2^{I(\Phi; \mathcal{F})}}{P}.$$

Therefore, a necessary condition for better uniqueness is using informative triggers. We begin with the conditional mutual information of the $n$-th trigger under threat model $\mathbf{A}_\epsilon$.

**Definition 1.** *Let the threat model be $\mathbf{A}_\epsilon$, denote the mutual information of $\mathbf{t}_n$ conditioned on queried triggers $\mathbf{t}_1, \cdots, \mathbf{t}_{n-1}$ by:*

$$I_\epsilon\left(\mathbf{t}_n | \mathbf{t}_{1:(n-1)}\right) = H(\phi_n | \phi_1, \cdots, \phi_{n-1}) - H(\phi_n | \phi_1, \cdots, \phi_{n-1}, \mathcal{F}). \tag{4}$$

The mutual information of all triggers is decomposed as:

$$I(\Phi; \mathcal{F}) = H(\Phi) - H(\Phi | \mathcal{F}) = \sum_{n=1}^{N} I_\epsilon\left(\mathbf{t}_n | \mathbf{t}_{1:(n-1)}\right). \tag{5}$$

In the vanilla setting, the verifier considers $\epsilon = 0$ and the modified model is always recognized as irrelevant from the original version. So $H(\phi_n|\phi_1, \cdots, \phi_{n-1}, \mathcal{F}) = 0$ and:

$$I_0\left(\mathbf{t}_n|\mathbf{t}_{1:(n-1)}\right) = H(\phi_n|\phi_1, \cdots, \phi_{n-1}). \tag{6}$$

The complexity in computing the r.h.s. of Eq. (6) by iterating over all possible values of $(\phi_1, \cdots, \phi_{n-1}, \phi_n)$ is of order $\mathcal{O}(C^n)$. Instead, we resort to Algo. 1 where all registered models are segmented into disjoint sets according to their partial fingerprints on the first $(n-1)$ triggers. Fingerprints that never appear are ignored. The complexity is reduced to $\mathcal{O}(\max\{P^2 n, PC\})$ and is acceptable since $P \ll C^n$ usually holds in practice, especially when $n$ is large.

In non-trivial cases where the tolerance $\epsilon > 0$, the verifier attributes modified models as copies of their original version and $H(\phi_n|\phi_1, \cdots, \phi_{n-1}, \mathcal{F})$ no longer always equals zero. A lower bound of the conditional mutual information of each trigger is given by the following theorem.

**Theorem 1.** *Let the threat model be $\mathbf{A}_\epsilon$ in Eq. (1), then:*

$$I_\epsilon\left(\mathbf{t}_n|\mathbf{t}_{1:(n-1)}\right) \geq I_0\left(\mathbf{t}_n|\mathbf{t}_{1:(n-1)}\right) - h(\epsilon). \tag{7}$$

*where $h(\epsilon) = -\delta_\mathcal{T}(\epsilon)\log_2 \delta_\mathcal{T}(\epsilon) - (1 - \delta_\mathcal{T}(\epsilon))\log_2(1 - \delta_\mathcal{T}(\epsilon)) + \delta_\mathcal{T}(\epsilon)\log_2(C - 1)$.*

Conceptually, Theorem 1 is a variant of Fano's inequality. The complete proof is given in Appendix A.

## 3.2 Greedy Optimization of Triggers

Being equipped with the conditional mutual information of each trigger, we proceed to optimize the collection of triggers. Given the budget $\hat{N} \leq N$, the optimized collection of triggers $\hat{\mathbf{T}}$ with size $\hat{N}$ is selected by Algo. 2 where the trigger with the largest conditional entropy is iteratively included. Remarkably, $\hat{\mathbf{T}}$ is a permutation of a subset of $\mathbf{T}$.

Theorem 1 indicates that even for $\epsilon > 0$, the mutual information provided by the $n$-th trigger is lower bounded by $I_0$ minus a constant. So the cumulative mutual information provided by triggers selected according to Algo. 2 is lower bounded under arbitrary threat models.

Selecting a fixed number of triggers that provides the largest mutual information is essentially a combinatorial optimization problem that is NP-hard. We prove that our strategy selects triggers whose mutual information is lower bounded compared with the optimal triggers in theory. Let $\underline{I}_\epsilon\left(\mathbf{t}_n|\mathbf{t}_{1:(n-1)}\right) = \max\left\{0, I_0\left(\mathbf{t}_n|\mathbf{t}_{1:(n-1)}\right) - h(\epsilon)\right\}$.

The mutual information provided by triggers $\mathbf{T}$ is no less than $g(\mathbf{T}) = \sum_{n=1}^N \underline{I}_\epsilon\left(\mathbf{t}_n|\mathbf{t}_{1:(n-1)}\right)$, which

---

**Algorithm 2** Greedily selecting $\hat{N}$ informative triggers.

---

**Input:** Budget $\hat{N}$, triggers $\mathbf{T}$, registered models $\mathbf{F}$
**Output:** A collection of triggers $\hat{\mathbf{T}}$, $|\hat{\mathbf{T}}| = \hat{N}$.
1: $\hat{\mathbf{T}} = \emptyset$
2: **for** $n = 1$ to $\hat{N}$ **do**
3:     $m = 0, \mathbf{r} \in \mathbf{T} \setminus \hat{\mathbf{T}}$
4:     **for** $\mathbf{t} \in \mathbf{T} \setminus \hat{\mathbf{T}}$ **do**
5:         **if** $I_0(\mathbf{t}|\hat{\mathbf{T}}) > m$ **then**
6:             $m = I_0(\mathbf{t}|\hat{\mathbf{T}}), \mathbf{r} = \mathbf{t}$
7:         **end if**
8:     **end for**
9:     $\hat{\mathbf{T}} = \hat{\mathbf{T}} \cup \{\hat{\mathbf{t}}_n = \mathbf{r}\}$
10: **end for**
11: **Return** $\hat{\mathbf{T}}$

---

turns out to be a non-negative, monotonically increasing, and submodular function in $\mathbf{T}$ [23–25]. Because for $\mathbf{V} \subset \mathbf{U} \subset \mathbf{T}$ and $\mathbf{t} \in \mathbf{T} \setminus \mathbf{U}$:

$$g(\mathbf{U} \cup \{\mathbf{t}\}) - g(\mathbf{U}) = \max\{0, H(\mathbf{t}|\mathbf{U}) - h(\epsilon)\}$$
$$\leq \max\{0, H(\mathbf{t}|\mathbf{V}) - h(\epsilon)\} = g(\mathbf{V} \cup \{\mathbf{t}\}) - g(\mathbf{V}).$$

The submodularity guarantees that the lower bound of mutual information given by $\hat{N}$ greedily selected triggers is no less than $\left(1 - \frac{1}{e}\right)$ of that of $\hat{N}$ optimal triggers in theory, i.e.,

$$\sum_{n=1}^{\hat{N}} \underline{I}_\epsilon\left(\hat{\mathbf{t}}_n|\hat{\mathbf{t}}_{1:(n-1)}\right) \geq \left(1 - \frac{1}{e}\right) \sum_{n=1}^{\hat{N}} \underline{I}_\epsilon\left(\tilde{\mathbf{t}}_n|\tilde{\mathbf{t}}_{1:(n-1)}\right). \tag{8}$$

where $\hat{\mathbf{t}}_n/\tilde{\mathbf{t}}_n$ is the $n$-th trigger in the greedily selected/optimal collection with size $\hat{N}$.

In summary, When the verifier can only query $\hat{N}$ instead of all $N$ triggers, we recommend retrieving the fingerprint with $\hat{\mathbf{T}}$ instead of $\hat{N}$ random triggers in $\mathbf{T}$. This discussion further implies a necessary condition on the number of triggers.

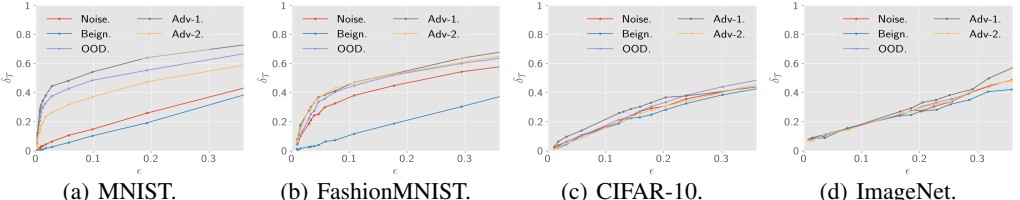

|       (a) MNIST. | (b) FashionMNIST. | (c) CIFAR-10. | (d) ImageNet. |

Figure 3: The robustness of studied DNN fingerprints measured in $\delta_{\mathcal{T}}(\epsilon)$, averaged on 600 models under the fine-pruning attack.

**Theorem 2.** *Let the threat model be* $\mathbf{A}_\epsilon$, *a necessary condition for the number of triggers is:*

$$N \geq \min \left\{ \hat{N} : \sum_{n=1}^{\hat{N}} \underline{I}_\epsilon \left( \hat{\mathbf{t}}_n | \hat{\mathbf{t}}_{1:(n-1)} \right) \geq \left( 1 - \frac{1}{e} \right) \log_2 P \right\}. \tag{9}$$

*If $\epsilon = 0$ and $N$ fails to meet Eq. (9) then any collection of $N$ triggers cannot trace the copyright of all registered models. If $\epsilon > 0$ and $N$ fails to meet Eq. (9) then any collection of $N$ triggers has a risk of failing to trace the copyright of all registered models.*

*Proof.* If $N$ fails to satisfy Eq. (9) then the optimal $N$ triggers might provide less information than $\log_2 P$ bits due to Eq. (8):

$$\sum_{n=1}^{N} \underline{I}_\epsilon \left( \tilde{\mathbf{t}}_n | \tilde{\mathbf{t}}_{1:(n-1)} \right) \leq \frac{1}{1 - \frac{1}{e}} \sum_{n=1}^{N} \underline{I}_\epsilon \left( \hat{\mathbf{t}}_n | \hat{\mathbf{t}}_{1:(n-1)} \right) < \log_2 P.$$

When $\epsilon = 0$ we have $I_0 = \underline{I}_0$, so failing to satisfy Eq. (9) implies a deterministic failure in copyright tracing. Otherwise, we can only assert that copyright tracing has a chance to fail since the lower bound of mutual information provided by any combination of triggers is insufficient to identify all registered models. $\square$

## 3.3   Remarks

We make three remarks regarding the implications and applications of our analyses.

**Remark 1: Optimizing the distribution of triggers.** Theorem 2 gives two directions to reduce the number of triggers for copyright tracing. The first is to reduce $\delta_{\mathcal{T}}(\epsilon)$, i.e., to increase the robustness, since $h(\epsilon)$ monotonically increases in $\delta_{\mathcal{T}}(\epsilon)$ when $\delta_{\mathcal{T}}(\epsilon)$ is small. The second is to increase $I_0 \left( \mathbf{t}_n | \mathbf{t}_{1:(n-1)} \right)$. Unfortunately, neither direction can be directly transformed into changes to $\mathcal{T}$. Moreover, there might be a trade-off between robustness and mutual information or uniqueness according to our empirical studies presented in the next section.

**Remark 2: Fingerprint vs. watermark.** In contrast to fingerprinting triggers, watermarking triggers have labels that are deliberately assigned. It can always be expected that $I_0 \left( \mathbf{t}_n | \mathbf{t}_{1:(n-1)} \right) = \log_2 C$ for watermarking triggers (unless $n$ is so large that queried triggers have already provided enough information for copyright tracing). Despite this advantage, fingerprinting schemes are applicable in more settings so they remain a competitive option.

**Remark 3: Generalization to non-classifiers.** Our method can be generalized to black-box copyright tracing systems for non-classifiers such as multimedia content generators [26]. The bridge is considering the basic copyright interpreter as a classifier [27, 28]. An example on a copyright tracing system for generative language models is given in Appendix B.

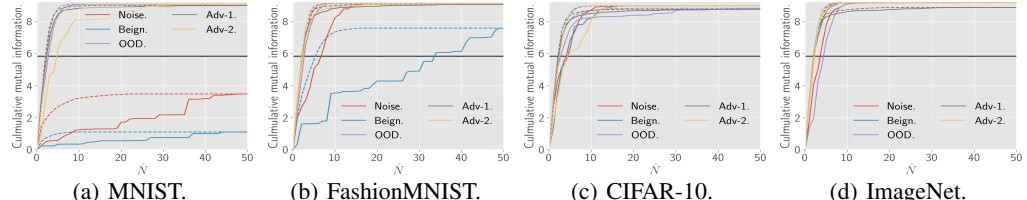

Figure 4: The cumulative mutual information (in bit) provided by triggers in the original order (the solid curves) and triggers selected by the greedy algorithm (the dashed curves) when $\epsilon = 0$. The black line marks $\left(1 - \frac{1}{e}\right) \log_2 600$ bits.

## 4 Experiments and Results

### 4.1 Settings

Following the settings of existing studies [29], we conducted experiments on four classification tasks: MNIST [30], FashionMNIST [31], CIFAR-10 [32], and ImageNet [33]. The number of classes was $C = 10$ (randomly drawn from ImageNet). A series of DNN models were trained as registered models. The sources of heterogeneity were (I) Four network architectures including LeNet-5 [34], VGG-16 [35], ResNet-18, and ResNet-34 [36]. (II) Five learning rates ranging from 0.02 to 0.1. (III) Two learning schedules with step lengths 5 and 10. (IV) Five training epochs ranging from 10 to 60. (V) Three random downsampling of training data. So, there were $P = 600$ models for each task. We used four GeForce RTX 2080 Ti GPUs for acceleration. All experiments were implemented using the PyTorch framework. Link to the code repo is given in the Appendix C.

We considered five representative DNN fingerprinting schemes. For each task, 50 triggers were produced by each scheme. The robustness was measured under the fine-pruning attack [20], results are shown in Fig. 3).

**(I) Noise** [9]. The pixel of each trigger was randomly generated from a normal distribution whose mean and variance were identical to samples from the training dataset. **(II) Benign** [4, 37, 38]. Each trigger was randomly drawn from the training dataset. **(III) OOD** [3, 10]. Triggers are randomly drawn across training datasets. For classifiers on MNIST and FasionMNIST, each trigger was randomly drawn from the training dataset of CIFAR-10. For classifiers on CIFAR-10 and ImageNet, triggers were drawn from MNIST. **(IV) Adv-1** [11, 13]. The first category of adversarial samples was produced from an ordinary SGD-based adversarial attack [39]. The victim model was a randomly chosen classifier. **(V) Adv-2**. The second category of adversarial samples was produced from the same SGD-based adversarial attack, but initial images were noises.

### 4.2 Results of Using Informative Triggers

#### 4.2.1 Baseline Setting: $\epsilon = 0$

The cumulative mutual information provided by triggers when $\epsilon = 0$ is visualized in Fig. 4. The necessary condition for the number of triggers given by Theorem 2 was uniformly no less than 4 in all combinations of task and fingerprinting scheme, i.e., it is impossible to trace all models with $\lceil \log_{10} 600 \rceil = 3$ triggers. Greedily selected triggers were always more informative than the original triggers. In all cases, the first 15 triggers selected by the greedy algorithm have provided the same amount of information as all 50 triggers, so extra querying is redundant.

#### 4.2.2 Adversarial Setting: $\epsilon > 0$

To delve into the influence of greedy selection on the uniqueness rate $u$ in adversarial environments, we computed $u$ defined by Eq. (3). Results are listed in Table 2. When the number of triggers is limited, using greedily selected triggers always yielded larger mutual information and secured the uniqueness of more models. This phenomenon appeared in all cases regardless of the task, the choice of fingerprinting scheme, the number of triggers, and the threat model with no exception. Notably,

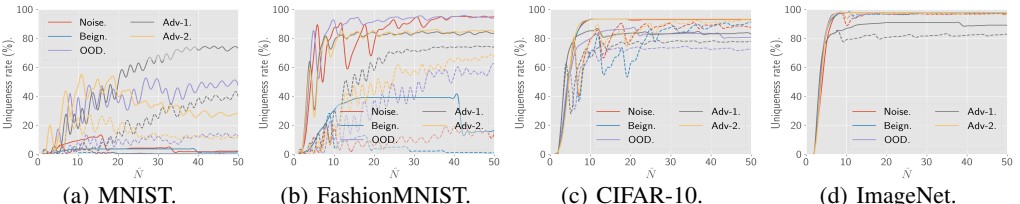

|  | (a) MNIST. | (b) FashionMNIST. | (c) CIFAR-10. | (d) ImageNet. |

Figure 5: Uniqueness rate (%) provided by greedily selected informative triggers when $\epsilon = 2.5\%$ (the solid curves) and $\epsilon = 10.0\%$ (the dashed curves).

Table 2: Uniqueness rate of registered models (%). For A/B in each entry, A is the uniqueness rate provided by the original order of triggers, and B is the uniqueness rate provided by greedily selected triggers. The dataset is MNIST, FashionMNIST, CIFAR-10, and ImageNet from top to bottom. The better scheme in each setting is highlighted in bold.

| $\epsilon$ | Noise $\hat{N}=5$ | Noise $\hat{N}=10$ | Noise $\hat{N}=15$ | Benign $\hat{N}=5$ | Benign $\hat{N}=10$ | Benign $\hat{N}=15$ | OOD $\hat{N}=5$ | OOD $\hat{N}=10$ | OOD $\hat{N}=15$ | Adv-1 $\hat{N}=5$ | Adv-1 $\hat{N}=10$ | Adv-1 $\hat{N}=15$ | Adv-2 $\hat{N}=5$ | Adv-2 $\hat{N}=10$ | Adv-2 $\hat{N}=15$ |
|---|---|---|---|---|---|---|---|---|---|---|---|---|---|---|---|
| 0.0 | 1.7/**5.0** | 3.0/**9.5** | 3.3/**12.2** | 0.5/**1.8** | 0.7/**3.7** | 2.0/**3.7** | 63.0/**78.2** | 93.7/**96.5** | 96.5/**96.5** | 69.5/**81.8** | 89.5/**91.5** | 90.8/**93.5** | 23.2/**58.7** | 62.5/**92.2** | 64.8/**95.3** |
| 5.0 | 1.7/**5.0** | 3.0/**9.5** | 3.3/**3.7** | 0.5/**1.8** | 0.7/**3.7** | 2.0/**3.7** | 0.5/**0.7** | 3.0/**7.2** | 7.0/**15.2** | 0.7/**0.8** | 8.2/**17.7** | 16.3/**19.6** | 3.8/**12.0** | 6.7/**25.2** | 7.8/**33.7** |
| 10.0 | 1.7/**5.0** | 0.8/**2.8** | 0.2/**3.7** | 0.5/**1.8** | 0.2/**1.2** | 0.5/**1.2** | 0.5/**0.7** | 3.0/**8.2** | 1.7/**4.7** | 0.7/**0.8** | 1.5/**2.8** | 3.7/**4.9** | 3.8/**12.0** | 6.7/**25.2** | 4.7/**20.5** |
| 0.0 | 14.3/**71.7** | 64.0/**96.0** | 90.0/**96.0** | 0.8/**8.5** | 2.5/**30.5** | 3.7/**38.2** | 54.0/**84.5** | 96.0/**96.0** | 96.0/**96.0** | 72.8/**81.5** | 87.8/**92.3** | 90.8/**94.2** | 71.5/**80.0** | 88.8/**95.5** | 83.7/**91.0** |
| 5.0 | 1.5/**12.0** | 7.5/**52.8** | 7.7/**15.8** | 0.8/**8.5** | 2.5/**30.5** | 3.7/**38.2** | 10.2/**21.3** | 29.5/**43.2** | 54.7/**54.7** | 28.5/**30.8** | 57.5/**63.7** | 68.5/**71.5** | 22.2/**26.0** | 43.0/**54.2** | 59.5/**71.5** |
| 10.0 | 1.5/**12.0** | 1.2/**17.8** | 2.3/**5.7** | 0.8/**8.5** | 0.7/**5.8** | 0.7/**10.7** | 0.2/**0.8** | 4.7/**9.8** | 26.7/**26.7** | 0.5/**1.2** | 25.0/**30.0** | 55.7/**60.0** | 0.5/**0.5** | 16.3/**20.5** | 38.7/**47.5** |
| 0.0 | 22.0/**59.8** | 76.8/**93.0** | 91.3/**93.3** | 17.0/**62.0** | 56.3/**93.3** | 89.8/**93.3** | 47.3/**56.8** | 77.3/**80.0** | 78.8/**85.8** | 65.2/**74.5** | 82.0/**85.3** | 84.2/**87.7** | 17.7/**61.2** | 53.2/**93.0** | 83.8/**93.3** |
| 5.0 | 22.0/**59.8** | 76.8/**93.0** | 79.2/**91.5** | 17.0/**62.0** | 56.3/**93.3** | 69.5/**84.7** | 47.3/**56.8** | 77.3/**80.0** | 74.2/**76.2** | 65.2/**74.5** | 77.7/**78.3** | 80.8/**81.7** | 17.7/**61.2** | 53.2/**93.0** | 53.0/**92.0** |
| 10.0 | 22.0/**59.8** | 36.2/**79.7** | 48.3/**80.0** | 17.0/**62.0** | 22.2/**71.3** | 42.8/**60.3** | 47.3/**56.8** | 69.3/**69.8** | 72.0/**72.5** | 18.0/**29.2** | 67.7/**69.0** | 74.8/**75.4** | 17.7/**61.2** | 18.7/**77.8** | 28.5/**80.5** |
| 0.0 | 40.8/**76.0** | 91.3/**97.7** | 97.5/**98.0** | 66.2/**83.7** | 97.7/**98.0** | 97.8/**98.0** | 29.5/**86.0** | 91.5/**97.7** | 97.3/**98.0** | 74.7/**78.3** | 84.8/**87.0** | 87.0/**90.0** | 64.8/**84.0** | 97.5/**98.0** | 98.0/**98.0** |
| 5.0 | 40.8/**76.0** | 91.3/**97.7** | 97.5/**98.0** | 66.2/**83.7** | 97.7/**98.0** | 97.8/**98.0** | 29.5/**86.0** | 91.5/**97.7** | 97.3/**98.0** | 74.7/**78.3** | 84.8/**87.0** | 87.0/**90.0** | 64.8/**84.0** | 97.5/**98.0** | 98.0/**98.0** |
| 0.0 | 40.8/**76.0** | 74.3/**89.2** | 96.2/**97.0** | 66.2/**83.7** | 97.7/**98.0** | 91.7/**96.7** | 29.5/**86.0** | 69.2/**94.2** | 95.0/**96.8** | 74.7/**78.3** | 78.8/**79.2** | 82.7/**83.0** | 64.8/**84.0** | 89.5/**93.5** | 96.5/**97.8** |

although the necessary condition in Theorem 2 is not a guarantee of perfect uniqueness, satisfying Eq. (9) implies a minimal uniqueness rate of at least 56.8%.

Table 2 also indicates that fewer models' fingerprints could remain unique when the threat model became stronger, i.e., when $\epsilon$ increased. One interesting finding is that some fingerprints might not have a higher uniqueness rate when the number of triggers increased. We suspect that when the threat model becomes very strong, the incremental mutual information for a large $\hat{N}$ can be completely nullified, as what is implied by the form of the lower bound given in Theorem 1. To examine this phenomenon, we recorded the uniqueness rate w.r.t. $\hat{N}$ under different threat models. Results shown in Fig. 5 demonstrate that the uniqueness rate was not always increased in $\hat{N}$. The fluctuations are due to rounding issues in computing the uniqueness rate, where only two fingerprints with at most $\lceil \delta_{\mathcal{T}}(\epsilon)\hat{N} \rceil$ differences were recognized as the same.

### 4.2.3 Re-evaluation of DNN Fingerprinting Schemes

In addition to robustness, statistics in Table 2 and Fig. 5 reveal a different ranking among examined DNN fingerprinting schemes. For example, despite the robustness, both **Noise** and **Benign** failed in MNIST by providing the least information, so did **Benign** in FashionMNIST. The reason is that triggers similar to normal samples are robust due to their entanglement with the classifier's functionality. However, this similarity means models trained on similar data perform alike, offering less model-specific information, especially for simple, high-accuracy tasks. **Adv-2** appeared to be the optimal choices in CIFAR-10 when $\hat{N} \leq 15$ regarding the uniqueness rate. This result is non-trivial since **Adv-2** was not judged as the most informative scheme before applying the greedy algorithm as shown in Fig. 4(c). For ImageNet, **Adv-1** outperformed **Noise** and **OOD** in their original setting when $N$ is small. However, both **Noise** and **OOD** provided perfect uniqueness after the greedy selection process, while **Adv-1** failed. We emphasize that existing robustness-oriented evaluation of DNN fingerprinting schemes tends to overlook these relationships between robustness and uniqueness and overfits oversimplified settings where $P$ is very small and/or $N/\hat{N}$ is very large.

The difficulty in tracing the copyright of DNN models also varies with the task. The more difficult the task is, the easier it is to distinguish models, since models trained on complex datasets tend to be more diversified instead of overfitting a local optimum.

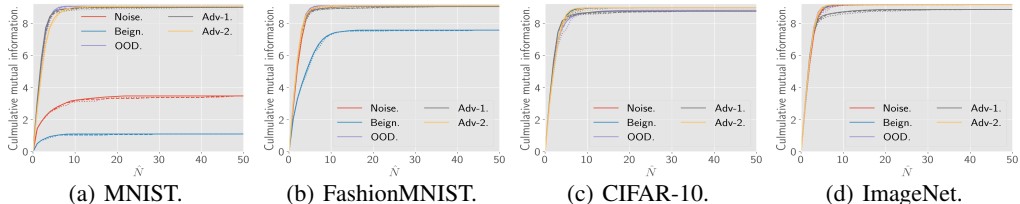

|  | (a) MNIST. | (b) FashionMNIST. | (c) CIFAR-10. | (d) ImageNet. |

Figure 6: The culmulative mutual information (in bit) provided by greedily selected triggers with 100 models (the dotted curves), 200 models (the dashed curves), and 600 models (the solid curves)..

Table 3: Uniqueness rate within all 600 models (%). The dataset is MNIST, FashionMNIST, CIFAR-10, and ImageNet from top to bottom. R denotes randomly drawn triggers, 100/200/600 denotes the number of models used to greedily select triggers (i.e., the size of $\mathbf{F}$ in Algo. 1). Settings that satisfy the order $R \leq 100 \leq 200 \leq 600$ are highlighted in green.

| | OOD | | | | | | | | | | | | Adv-2 | | | | | | | | | | | |
| | $\hat{N}=5$ | | | | $\hat{N}=10$ | | | | $\hat{N}=15$ | | | | $\hat{N}=5$ | | | | $\hat{N}=10$ | | | | $\hat{N}=15$ | | | |
| $\epsilon$ | R | 100 | 200 | 600 | R | 100 | 200 | 600 | R | 100 | 200 | 600 | R | 100 | 200 | 600 | R | 100 | 200 | 600 | R | 100 | 200 | 600 |
|---|---|---|---|---|---|---|---|---|---|---|---|---|---|---|---|---|---|---|---|---|---|---|---|---|
| 0.0 | 63.0 | 71.8 | 73.2 | 78.2 | 93.7 | 96.2 | 96.2 | 96.5 | 96.5 | 96.5 | 96.5 | 96.5 | 23.2 | 50.2 | 59.2 | 58.7 | 62.5 | 83.3 | 91.5 | 92.2 | 64.8 | 92.0 | 93.8 | 95.3 |
| 5.0 | 0.5 | 0.5 | 1.0 | 0.7 | 3.0 | 8.0 | 8.0 | 8.2 | 7.0 | 10.7 | 14.2 | 15.2 | 3.8 | 8.5 | 11.2 | 12.0 | 6.7 | 7.3 | 20.2 | 25.2 | 7.8 | 24.7 | 28.3 | 33.7 |
| 10.0 | 0.5 | 0.5 | 1.0 | 0.7 | 3.0 | 8.0 | 8.0 | 8.2 | 1.7 | 4.2 | 5.2 | 4.7 | 3.8 | 8.5 | 11.2 | 12.0 | 6.7 | 7.3 | 20.2 | 25.2 | 4.7 | 13.0 | 16.7 | 20.5 |
| 0.0 | 54.0 | 76.2 | 79.3 | 84.5 | 96.0 | 96.0 | 96.0 | 96.0 | 96.0 | 96.0 | 96.0 | 96.0 | 71.5 | 77.2 | 74.5 | 80.0 | 88.8 | 91.0 | 94.2 | 95.5 | 83.7 | 94.2 | 94.8 | 91.0 |
| 5.0 | 10.2 | 16.3 | 20.8 | 21.3 | 29.5 | 34.5 | 37.0 | 43.2 | 54.7 | 54.7 | 54.7 | 54.7 | 22.2 | 24.2 | 24.8 | 26.0 | 43.0 | 49.5 | 52.2 | 54.2 | 59.5 | 59.7 | 60.3 | 71.5 |
| 10.0 | 0.2 | 0.5 | 1.0 | 0.8 | 4.7 | 5.0 | 8.2 | 9.8 | 26.7 | 26.7 | 26.7 | 26.7 | 0.5 | 0.5 | 0.5 | 0.5 | 16.3 | 18.3 | 18.7 | 20.5 | 38.7 | 39.7 | 40.0 | 47.5 |
| 0.0 | 47.3 | 55.0 | 55.8 | 56.8 | 77.3 | 77.8 | 79.8 | 80.0 | 78.8 | 83.7 | 85.2 | 85.8 | 17.7 | 47.0 | 55.8 | 61.2 | 53.2 | 82.7 | 91.7 | 93.0 | 83.8 | 93.0 | 93.0 | 93.3 |
| 5.0 | 47.3 | 55.0 | 55.8 | 56.8 | 77.3 | 77.8 | 79.8 | 80.0 | 74.2 | 75.8 | 76.0 | 76.2 | 17.7 | 47.0 | 55.8 | 61.2 | 53.2 | 82.7 | 91.7 | 92.0 | 53.0 | 84.2 | 86.2 | 92.0 |
| 10.0 | 47.3 | 55.0 | 55.8 | 56.8 | 69.3 | 69.7 | 69.7 | 69.8 | 72.0 | 72.0 | 72.3 | 72.5 | 17.7 | 47.0 | 55.8 | 61.2 | 18.7 | 44.3 | 69.3 | 77.8 | 28.5 | 51.8 | 55.7 | 80.5 |
| 0.0 | 29.5 | 65.2 | 82.3 | 86.0 | 91.5 | 94.8 | 95.8 | 97.7 | 97.3 | 97.5 | 97.5 | 98.0 | 64.8 | 79.2 | 83.3 | 84.0 | 97.5 | 97.7 | 97.9 | 98.0 | 98.0 | 98.0 | 98.0 | 98.0 |
| 5.0 | 29.5 | 65.2 | 82.3 | 86.0 | 91.5 | 94.8 | 95.8 | 97.7 | 97.3 | 97.5 | 97.5 | 98.0 | 64.8 | 79.2 | 83.3 | 84.0 | 97.5 | 97.7 | 97.9 | 98.0 | 98.0 | 98.0 | 98.0 | 98.0 |
| 10.0 | 29.5 | 65.2 | 82.3 | 86.0 | 69.2 | 73.7 | 85.0 | 94.2 | 95.0 | 96.5 | 96.5 | 96.8 | 64.8 | 79.2 | 83.3 | 84.0 | 89.5 | 93.8 | 94.2 | 93.5 | 96.5 | 97.2 | 97.3 | 97.8 |

## 4.3 Scalability to Online Copyright Tracing

An final concern is whether greedily selected triggers remain informative when more models are registered online. Specifically, we assume that the verifier only obtains a small number of DNN models at the beginning, with which he/she greedily selects a series of triggers. We are interested in whether these triggers still outperform randomly drawn triggers on unseen models or not.

The greedily selected triggers according to either 100 or 200 models performed almost identically to those selected according to all 600 models regarding the culmulative information, as shown in Fig. 6. The failure to refer to all registered DNN models had a small impact on the uniqueness rate as shown in Table 3. In almost all cases, the greedily selected triggers outperformed randomly selected triggers, even if only 100 models were considered. In general, the more models verifier can observe during greedy selection process, the larger the uniqueness rate is. This fact is justified by the dominance of green entries in Table 3. Therefore, we recommend that the copyright verifier collects as many models as possible to select informative triggers, or choose to update the collection of triggers when the number of registered DNN models increases. Even if the number of observed models is small, the greedily selected triggers are more informative than randomly drawn triggers and result in a larger uniqueness rate.

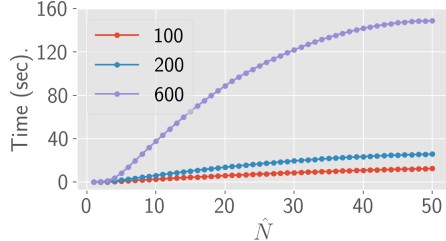

Figure 7: The time consumption of greedy trigger selection.

## 4.4 Overhead

We also evaluated the overhead of our method. The time consumption is no more than 160 seconds even when all 600 models were used to optimize 50 triggers (it has been shown that 15-30 triggers

were sufficient for copyright tracing), as shown in Fig. 7. This overhead is acceptable and is independent from the dataset.

## 5 Discussions

The fingerprint optimization algorithm proposed in this paper adopts informative triggers during the fingerprint selection phase, aiming to achieve enhanced uniqueness. It does not involve the design or implementation of the fingerprint scheme itself, and can be seamlessly integrated with trigger set-based fingerprint schemes to improve their effectiveness in practical application scenarios.

**Limitations.** More complex task scenarios beyond text generation, such as image and video generation, have not yet been discussed, which will also be the focus of our future work.

## 6 Conclusions

This paper explores uniqueness, a less frequently studied yet important dimension in evaluating DNN fingerprinting schemes for copyright tracing. After highlighting the significance and challenges regarding this property, we adopt an information theoretical perspective to quantify the contribution of each fingerprinting trigger. We design an algorithm to efficiently estimate the conditional mutual information of each trigger and propose a greedy algorithm that facilitates the efficiency of copyright tracing. Extensive experiments show that our method can be easily combined with arbitrary DNN fingerprinting schemes to improve the performance regarding uniqueness, even in the online setting. Our studies reveal several new insights in evaluating and comparing DNN fingerprinting schemes and suggest more attentions on uniqueness in addition to robustness.

## Acknowledgments and Disclosure of Funding

The work described in this paper was supported in part by the National Natural Science Foundation of China (62271307).

The AI-driven experiments, simulations and model training were performed on the robotic AI-Scientist platform of Chinese Academy of Sciences.

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

# A  Proof of Theorem 1

*Proof.* Consider an attack that changes each position of the fingerprint with a probability $\delta$. This attack is featured by a $C$-ary code of length $N$, among which $(1 - \delta)N$ positions equal zero to represent triggers whose predictions remain invariable. The other $\delta N$ positions are distributed in $\{1, 2, \cdots, C - 1\}$ to represent that the corresponding predictions have been shifted to the next $1, 2, \cdots, (C - 1)$-th class (modulo $C$).

We use $\phi_n^0$ to denote the output of the suspicious model on $\mathbf{t}_n$ before being attacked. For the first term in Eq. (4), :

$$
\begin{aligned}
H(\phi_n|\phi_1, \cdots, \phi_{n-1}) &\overset{(a)}{\geq} H(\phi_n|\phi_1, \cdots, \phi_{n-1}, \mathcal{A}) \\
&\overset{(b)}{=} H(\phi_n^0|\phi_1, \cdots, \phi_{n-1}, \mathcal{A}) \overset{(c)}{=} I_0\left(\mathbf{t}_n|\mathbf{t}_{1:(n-1)}\right).
\end{aligned}
\tag{10}
$$

in which *(a)* follows the basic properties of entropy, *(b)* and *(c)* hold since once the attack is known, the entire case can be reduced to the vanilla setting as if no attack has been applied.

The second term in Eq. (4) equals:

$$
\begin{aligned}
&H(\phi_n, \mathcal{A}_{1:n}|\phi_1, \cdots, \phi_{n-1}, \mathcal{F}) - H(\mathcal{A}_{1:n}|\phi_1, \cdots, \phi_n, \mathcal{F}) \\
&\overset{(a)}{=} H(\phi_n, \mathcal{A}_{1:n}|\phi_1, \cdots, \phi_{n-1}, \mathcal{F}) \\
&\overset{(b)}{=} H(\mathcal{A}_{1:n}|\phi_1, \cdots, \phi_{n-1}, \mathcal{F}) \overset{(c)}{=} H(\mathcal{A}_n) \\
&\overset{(d)}{\leq} -\delta \log_2 \delta - (1 - \delta)\log_2(1 - \delta) + \delta \log_2(C - 1),
\end{aligned}
\tag{11}
$$

where $\mathcal{A}_{1:n}$ and $\mathcal{A}_n$ denote the attack on corresponding triggers. All *(a)-(d)* use the attack's representation, *(b)* also relies on the chain rule of entropy. Combining Eq. (10) and Eq. (11) yields:

$$
\begin{aligned}
I_\epsilon\left(\mathbf{t}_n|\mathbf{t}_{1:(n-1)}\right) \geq &I_0\left(\mathbf{t}_n|\mathbf{t}_{1:(n-1)}\right) + \delta \log_2 \delta \\
&+ (1 - \delta)\log_2(1 - \delta) - \delta \log_2(C - 1).
\end{aligned}
\tag{12}
$$

The r.h.s. of Eq. (12) monotonically decreases in $\delta$ (when $\delta \leq 0.5$). Combining this observation with Eq. (2) yields Eq. (7). $\qquad\square$

# B  Applications on Copyright Tracing of Generative Language Models

We demonstrate the application of the proposed method on the copyright tracing of generative language models.

We built a collection of $P = 50$ generative language models including `GPT-2` [40] with 10 fine-tuned versions, `GPT-Neo-125M` [41] with 10 fine-tuned versions, `OPT-125M` [42] with 10 fine-tuned versions, `OPT-350M` with 10 fine-tuned versions, `Pythia-70M` [43], `Pythia-160M`, `Pythia-160M-deduped`, `T5-Small` [44], `Flan-T5-Small` [45], and `BLOOM-560M` [46]. Each fine-tuned version used one corpus from CMV [47], Yelp [48], TLDR [49], XSum [49], ELI5 [50], WP [51], ROC [52], HellaSwag [53], SQuAD [54], and SciGen [55].

Candidate basic copyright tracing algorithms were T5-Sentinel [56] ($C = 5$), T5-Hidden ($C = 5$), and LLMDet [57] ($C = 9$). Given a series of prompts, the suspicious model generates a series of texts, which are fed into the basic copyright tracing algorithm. The fingerprint of the suspicious model is the list of outputs from the copyright tracing algorithm. For example, LLMDet has captured texts generated from {0:Human, 1:GPT-2, 2:OPT, 3:UniLM, 4:LLaMA, 5:BART, 6:T5, 7:BLOOM, 8:GPT-Neo}. Given a list of seven prompts, `T5-Small` returns six sentences, which might be classified by LLMDet into [Human,GPT-2,T5,LLaMA,Human,T5,GPT-2], so the fingerprint of `T5-Small` under LLMDet can be encoded as [0,1,6,4,0,6,1], which can be interpreted as the fingerprint of a nine-class classifier.

Initially, $N = 200$ prompts were random drawn from the union of 10 corpura for fine-tuning. The culmulative mutual information provided by fingerprints from three algorithms is visualuzed in Fig. 8. It turns out that either T5-Sentinel or T5-Hidden was capable of distinguishing all 50 models with five prompts, although they were trained on only five models (Human, GPT3.5, PaLM, LLaMA, and GPT2-XL). Meanwhile, LLMDet has learned data generated from nine different models, but it failed

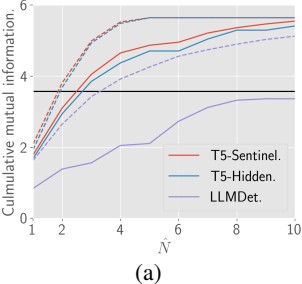 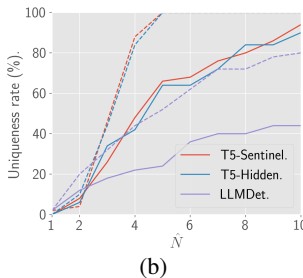

(a)                        (b)

Figure 8: (a) The culmulative mutual information (in bit) and (b) the uniqueness rate provided by prompts in the original order (the solid curves) and prompts seleted by the greedy algorithm (the dashed curves). The black line marks $\left(1 - \frac{1}{e}\right) \log_2 50$ bits.

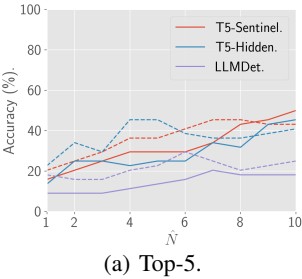 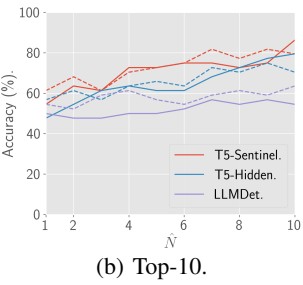

(a) Top-5.                           (b) Top-10.

Figure 9: The copyright tracing accuracy provided by prompts in the original order (the solid curves) and prompts seleted by the greedy algorithm (the dashed curves).

to differentiate all models even with ten prompts. After all, all three schemes' performance was boosted under the greedy selection framework.

The uniqueness declined sharply in the adversarial setting where we implemented fine-tuning with Wikimedia corpus [2] as adversarial modifications. Greedily selected triggers outperformed the baseline random setting as well. This is reflected in the copyright tracing accuracy in Fig. 9.

We remark that for generative language models, the conditional mutual information of a prompt depends on both the prompt's source corpus and the classifier algorithm (there is no extra copyright tracing classifier for DNN classifiers to be protected), our experiments suggested that the first factor also had a small influence as shown in Fig. 10. Although T5-Sentinal and T5-Hidden performed differently across corpura (using random prompts from a corpus), their performed almost identically after incorporating the greedy selection scheme. Meanwhile, the culmulative mutual information of LLMDet remained the lowest in all cases, yet our greedy selection scheme uniformly boosted its performance.

To simplify the evaluation, the random seeds within language models were manually fixed. In practice, the verifier is encouraged to feed a prompt to a language model for multiple times, record the predictions returned from the basic copyright tracing algorithm, and conduct a voting. It can be proven that when the error of this estimation for each prompt is bounded by $\epsilon$ (i.e., the probability that the prediction for this prompt differs from the statistical mode is no larger than $\epsilon$) then the bound in Eq. (8) should be relaxed into:

$$
\sum_{n=1}^{\hat{N}} \underline{I}_\epsilon\left(\hat{\mathbf{t}}_n | \hat{\mathbf{t}}_{1:(n-1)}\right) \geq
$$

$$
\left(1 - \frac{1}{e}\right)\left(\sum_{n=1}^{\hat{N}} \underline{I}_\epsilon\left(\tilde{\mathbf{t}}_n | \tilde{\mathbf{t}}_{1:(n-1)}\right) - \epsilon\hat{N}^2\left(1 - \frac{1}{\hat{N}}\right)\log_2 C\right) - \epsilon\hat{N}^2\log_2 C.
$$

(13)

---

[2]https://dumps .wikimedia.org.

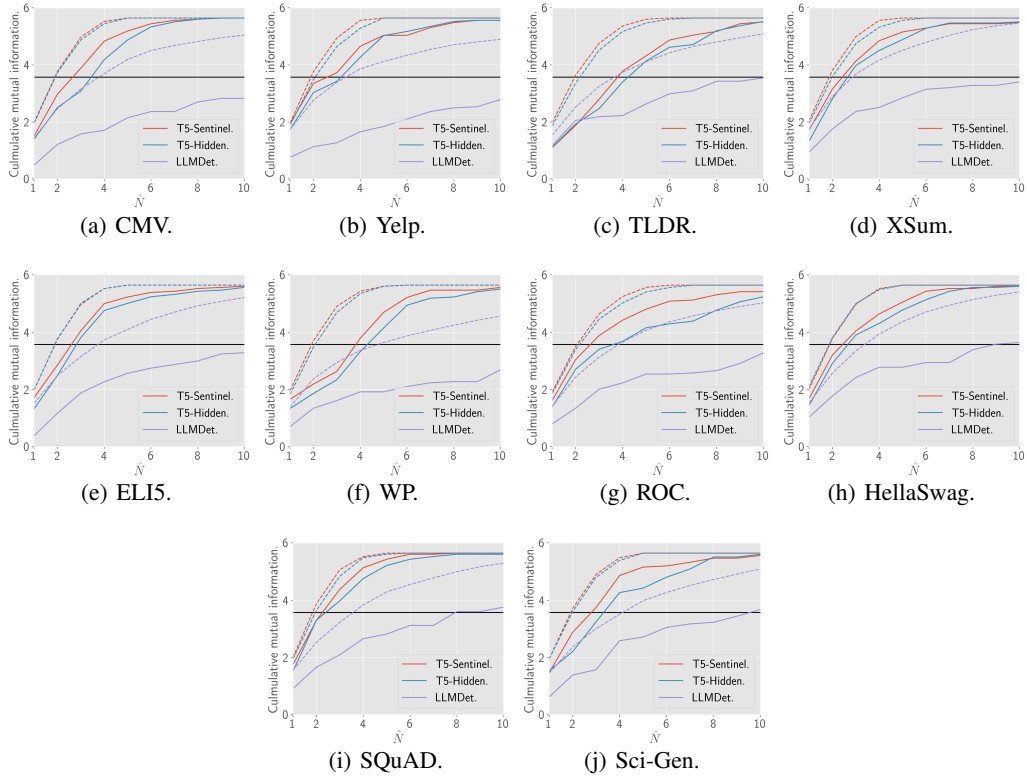

Figure 10: The culmulative mutual information (in bit) provided by prompts in the original order (the solid curves) and prompts seleted by the greedy algorithm (the dashed curves). The black line marks $\left(1 - \frac{1}{e}\right) \log_2 50$ bits.

The proof is similar to the induction in Lemma 2 in [58].

In conclusion, our scheme can be generalized to other non-classifiers and boost the performance of copyright tracing by increasing the uniqueness rate. Additionally, it can be used in the open setting where the models to be traced have not been included into the training set of basic copyright tracing algorithms, so it is necessary to use multiple prompts to extract their fingerprints.

# C   Code Repo Link

All codes for reproducibility in `https://github.com/zzmsmm/Informative_Triggers`.

