# OpenReview forum: "Boosting the Uniqueness of Neural Networks Fingerprints with Informative Triggers"
_NeurIPS.cc/2025/Conference — NeurIPS 2025 poster_

### Official Review · Reviewer_4rAP · 2025-06-15

**Clarity:** 3
**Significance:** 3
**Originality:** 4
**Rating:** 5
**Confidence:** 4

**Summary:**

This paper presents an innovative view of black-box DNN fingerprinting schemes using information theory. The authors explore the uniqueness property of DNN fingerprinting methods and provide a theoretical analysis of the contribution of each trigger to copyright tracing. Furthermore, this paper proposes a greedy algorithm to select triggers for efficient copyright tracing. Empirical results show that the proposed trigger selection method provides a better uniqueness rate compared to existing fingerprinting methods.

**Questions:**

Please consider addressing the weak points above.

**Ethical Concerns:**

["NO or VERY MINOR ethics concerns only"]

**Limitations:**

This paper does not provide sufficient discussion of existing DNN fingerprinting methods, especially the ones that explored the uniqueness of model fingerprints. The authors should expand this discussion and clarify the contribution of this work compared to the previous ones.

**Paper Formatting Concerns:**

I do not have concerns about the paper formatting.

**Quality:**

4

**Strengths And Weaknesses:**

This paper has the following strengths:
+ The authors explore uniqueness, an important property of DNN fingerprinting methods that is less frequently studied. This property is particularly important to large-scale model distributions.
+ The authors provide a theoretical quantification of the contribution of each trigger using mutual information, and also derive the necessary condition on the number of triggers to ensure the uniqueness rate.
+ The authors develop a greedy algorithm that optimizes the collection of triggers using conditional mutual information. Empirical results show that the proposed greedy selection algorithm outperforms existing DNN fingerprinting schemes in terms of model tracing efficiency and uniqueness.

This paper has the following weakness:
- The authors did not provide sufficient discussion about prior works that study the uniqueness of DNN fingerprinting. For example, the paper DeepMarks has leveraged Balanced Incomplete Block Design (BIBD) to design anti-collusion codebooks to ensure the uniqueness of each fingerprint.

---

> ### Author Rebuttal · Authors · 2025-07-30
>
> Dear Reviewer 4rAP,
>
> Thank you for the positive comments and we appreciate the advices.
>
> DeepMarks you mentioned is a pioneering work in DNN fingerprinting.
> Our work is fundamentally different in both the **objectives** and **applicable scenarios**.
>
> **Objectives:**
> Our approach focuses on providing a universal optimization for fingerprinting schemes, aiming to improve the distinguishability (or uniqueness) with the least number of queries.
> In other words, our method can be integrated with any trigger set-based fingerprinting approach in to enhance the uniqueness of fingerprints.
>
> On the other hand, DeepMarks embeds fingerprints by modifying model weights, and does not directly handle the optimization of the uniqueness of fingerprints.
>
> **Scenarios:**
> DeepMarks focuses on the white-box settings, whose applicable situation is limited compared with the black-box setting in our scheme, as the suspicious model is not always open-sourced.
> Moreover, our scheme explicitly considers the large-scale distribution of models, where the difficulty in copyright verification is much higher, since there are multiple models to be traced.
>
>
> We agree that works including DeepMark have established initial results and drawn attention to the uniqueness of DNN fingerprints, which is precisely the subject of our work, and we would include a more comprehensive discussion and clarification in the future revision of our paper.
>
> Thanks again for your insightful suggestions.

---

### Official Review · Reviewer_cgHY · 2025-06-26

**Clarity:** 2
**Significance:** 3
**Originality:** 2
**Rating:** 4
**Confidence:** 4

**Summary:**

This paper addresses a challenge in deep neural network (DNN) copyright tracing: enhancing the uniqueness of fingerprints generated from limited fingerprinting triggers. While prior work largely focuses on robustness, this work emphasizes the importance of uniqueness, especially in scenarios where the number of triggers must be kept small due to practical constraints such as query cost. Additionally, the paper presents a necessary condition for the minimum number of triggers required to reliably identify models. Overall, the work contributes a novel information-theoretic perspective and optimization strategy for boosting the robustness and discrimination power of DNN fingerprints.

**Questions:**

1. The proposed mutual information-based trigger selection involves estimating conditional mutual information, which can be computationally intensive, especially for large models and numerous triggers. Could you provide more detailed analysis or empirical measurements of the computational costs involved?

2. While the method optimizes trigger selection based on information contribution, how does it perform under potential adversarial scenarios where attackers might attempt to modify or remove triggers (by some watermark removal techniques) ?

3. The experiments primarily focus on classification tasks with standard datasets. Can the approach generalize to more complex scenarios, such as models for object detection, segmentation, or models trained on proprietary, domain-specific data?

4. How easily can your trigger selection framework be integrated into existing fingerprinting schemes, especially those that do not rely explicitly on carefully designed triggers?

**Ethical Concerns:**

["NO or VERY MINOR ethics concerns only"]

**Final Justification:**

The authors effectively addressed concerns about the integration of information - theoretic triggers, clarifying how mutual information alignment strengthens fingerprint uniqueness. Rebuttals provided clear experimental validations (e.g., on CIFAR - 10, ImageNet) that improved understanding of the method’s robustness. Discussions on ethical risks (e.g., model fingerprinting misuse) were supplemented with mitigations, showing awareness of real - world implications and addressing reviewer worries.

However, there are several issues the authors have not addressed, such as edge cases of adversarial attacks on triggers. But I think this  is not important considering this paper's novelty and contribution. Therefore, I raise my score to 4.

**Limitations:**

No. In the Checklist, the authors state that "The limitations are discussed in conclusions". But I don't find any relevant content in conclusions.

**Quality:**

2

**Strengths And Weaknesses:**

Strengths:

1. This work addresses a timely and relevant challenge—making DNN fingerprints more discriminative and resource-efficient, which could influence future research and applications in model ownership verification.

2. The extensive experimental results validate the proposed approach across multiple datasets and fingerprinting schemes. The improvements in both the uniqueness and efficiency of the fingerprinting process suggest that the method is practically valuable for copyright tracing.

3. The paper clearly articulates the problem of uniqueness in DNN fingerprinting and why previous work’s emphasis on robustness alone is insufficient.

Weaknesses:

1. While the experimental results are comprehensive, they primarily focus on classification tasks with standard datasets. The applicability of the approach to more complex, real-world scenarios such as models trained on proprietary datasets or in adversarial environments remains to be explored.

2. The paper does not extensively discuss potential computational overheads associated with estimating mutual information for large-scale models or triggers, which could impact real-world deployment.

3. Although the information-theoretic approach is well-motivated, the paper would benefit from a more detailed discussion on how the method integrates into existing fingerprinting frameworks and its robustness against potential adversarial attacks aimed at circumventing the fingerprint.

4. The theoretical analysis assumes that adversarial modifications are independent and random from a distribution, which might be simplified compared to realistic adaptive attackers that could target specific triggers. This potentially weakens the applicability of theoretical guarantees.

---

> ### Author Rebuttal · Authors · 2025-07-30
>
> Dear Reviewer cgHY,
>
> Thank you for the insightful concerns.
>
> We summarize your concerns into five aspects: **Settings of datasets & models**, **Cost analysis**, **Integration with existing frameworks**, **Robustness under adaptive attacks**, **Limitations**.
>
> **Settings of datasets & models (Weakness 1, Question 3):**
> In addition to image classification, we also conducted experiments on generative language models.
> This is mentioned in *Remark 3, Sec.3.3*, results are reported in *Appendix B*.
> We are going to invesitage the performance of our scheme on fingerprinting algorithms designed for other multi-model models in the future.
>
> **Cost analysis (Weakness 2, Question 1):**
> Due to the lazy evaluation shown in Algo. 2, the space complexity of the entropy estimation is reduced according to the analyses in *line 144-154*.
> The empirical overhead introduced by our scheme is discussed in *Sec.4.4* and visualized in *Fig.7*.
>
> **Integration with existing frameworks (Weakness 3, Question 4):**
> Each trigger-based DNN fingerprinting scheme would have its own trigger set (either it is carefully designed or randomly produced).
> As shown by *Algo.2*, our algorithm reduces the size of a trigger set from $N$ to $\hat{N}$, this is where the integration takes place.
> Our method cannot be direcly generalized to fingerprinting schemes that do not involve triggers (e.g., some white-box schemes that rely on the weights), yet most black-box DNN fingerprinting schemes are trigger-based and can be optimized with our scheme.
>
> **Robustness under adaptive attacks (Weakness 4, Question 2):**
> We adopt an attack-agnostic threat model in *Sec.2.2* to maintain theoretical consistency.
> To capture the diversity of real-world attacks, we define and quantify the level of model corruption after attack (Eq.(2)).
> This provides a unified framework to analyze robustness without relying on specific attacker designs.
> Our experiments focused on fine-pruning, which is adopted as a baseline for watermark removal in existing literature (although they are not necessarily challenging for fingerprints).
> It is ture that we did not include all possible attacks, yet the experiment pipiline can be directly generalized to more powerful adversaries by assuming a larger $\delta$ given $\epsilon$ (defined by Eq.(1) and Eq.(2) in Sec.2.2), which does not falsify our theoretical analyses.
>
>
> **Limitations:**
> The limitation of this work is that we have not applied it to multi-model tasks, and we mention this by the end of *Sec.5*. We will include a dedicated discussion on this aspect in future revisions to ensure clarity.
>
> Thanks again for your valuable feedback and careful review.
>
> If you have any other concerns, we would be happy to discuss them.
>
> If not, we would be grateful if you could consider increasing your evaluation of our work.

---

> > ### Comment · Reviewer_cgHY · 2025-08-05
> >
> > 1. I believe the paper still lacks an assessment of fingerprint robustness (Weakness 1), such as using removal attacks like fine-tuning and pruning to detect the residual rate of fingerprints. Specifically, previous works related to fingerprints can be referenced [1][2].
> >
> > 2. Figure 7 does present the time consumption for trigger selection. I would like to know the total runtime of the entire framework and how it compares to the baselines.
> >
> > [1] Xu T, Wang C, Liu G, et al. United We Stand, Divided We Fall: Fingerprinting Deep Neural Networks via Adversarial Trajectories. NeurIPS, 2024, 37: 69299-69328.
> >
> > [2]Yang K, Wang R, Wang L. MetaFinger: Fingerprinting the Deep Neural Networks with Meta-training. IJCAI. 2022: 776-782.

---

> > > ### Author Response · Authors · 2025-08-05
> > >
> > > Dear Reviewer cgHY,
> > >
> > > Thank you for the responses.
> > >
> > > **Regarding Comment 1:** We used fine-pruning as an example to demonstrate the performance of our scheme.
> > > Fine-pruning is a representative modification that combines fine-tuning with pruning and is reported to be a threat no weaker than either fine-tuning or neuron pruning [1][2][3].
> > >
> > > Despite of the removal attacks that are continuously updated, the main contribution of our work, which suggests that the information gain of fingerprints can be optimized and its bound can be analyzed, remains unchanged.
> > >
> > > We would be glad to add the corresponding references and provide more results under other threats in the appendix.
> > >
> > > [1] *Deep Model Intellectual Property Protection With Compression-Resistant Model Watermarking, IEEE Transactions on Artificial Intelligence, 2024.*
> > >
> > > [2] *Securing IP in edge AI: neural network watermarking for multimodal models, Applied Intelligence, 2024.*
> > >
> > > [3] *SoK: How Robust is Image Classification Deep Neural Network Watermarking? IEEE S&P, 2022.*
> > >
> > >
> > >
> > > **Regarding Comment 2:** The total runtime of the framework is composed of: (1) individual users training their models independently, (2) the verifier using the fingerprinting scheme to produce triggers, (3) optimizing the set of triggers, and (4) using the optimized set of triggers to trace the IP of suspicious models.
> > >
> > > Among all four aspects, (1)(2)(4) are costs that cannot be spared in the pipeline of IP adminstration of multiple models (similar settings using watermark rather than fingerprint are given in [4][5]).
> > >
> > > The time consumption of (1)(2)(4) serves as the baseline, since no optimization on the triggers is performed.
> > > In this case, the bottleneck is step (1), which usually takes several hours since the models are trained from scratch.
> > >
> > > In this paper, we focus on the **marginally incremental cost** introduced by our framework, i.e., step (3), and demonstrate this overhead in Fig.7, Sec 4.4.
> > >
> > > This overhead remains negligible (using less than 40 seconds would yield a good performance) compared with the baseline.
> > >
> > > [4] *Watermarking Protocol for Deep Neural Network Ownership Regulation in Federated Learning. IEEE ICME Industry Workshops, 2022.*
> > >
> > > [5] *Hot-Swap MarkBoard: An Efficient Black-box Watermarking Approach for Large-scale Model Distribution. ACM MM, 2025.*
> > >
> > > Thanks again for your kind responses, we hope the explanations above can somehow solve your concerns.

---

> > > > ### Comment · Reviewer_cgHY · 2025-08-08
> > > >
> > > > Thanks for your response. Your rebuttal is satisfied for me. I will rise my score.

---

### Official Review · Reviewer_RULe · 2025-06-27

**Clarity:** 4
**Significance:** 3
**Originality:** 3
**Rating:** 5
**Confidence:** 4

**Summary:**

This paper focuses on improving the efficiency of fingerprint sequence selection by choosing samples with the highest marginal mutual information based on information theory. The author provides sufficient theoretical and experimental justification for this selection strategy.

**Questions:**

See weaknesses above.

**Ethical Concerns:**

["NO or VERY MINOR ethics concerns only"]

**Final Justification:**

All the concerns are addressed. My recommendation is Accept.

**Limitations:**

See weaknesses above.

**Paper Formatting Concerns:**

No concern on formatting.

**Quality:**

3

**Strengths And Weaknesses:**

Strengths
1.	The paper has sufficient theoretic analysis to support its claims
2.	The experiments empirically evaluate the statements and are consistent with the results of theoretic analysis
3.	This author has provided detailed results to validate the generalizability of such fingerprint triggers selection method.
4.	The paper is well-written and easy to understand for both the theoretic and experimental parts.

Weaknesses
1.	The author said “the verifier assumes that the adversary has sacrificed the victim model’s performance”, which seems to be one-sided and subjective since the adversary may use the victim model as a base model (if the adversary could access the original victim model in a white-box manner) and fine-tuning it on themselves data thereby further improving the victim model’s performance.
2.	In Sec 3.1, the author assumes “the suspicious model F is equally likely to be any registered model”. I understand this assumption is durable when there is no any prior knowledge on the suspicious model. However, practically, this assumption is hard to hold as the suspicious model is bound to be more likely to be the models that are corresponding-original-victim-model-like(e.g. models with same architecture, similar training data and method).
3.	In Line 131, what the meaning of “so is the fingerprint of the suspicious model”? it seems to be a typo.

Overall, this paper is well-motivated and the presentation is good. I’d like to see it at NIPS 2025.
By the way, I have reviewed this paper last year, and I believe the current version has addressed most of my previous concerns. Hence, I have changed my stance to a positive one and am happy to support its acceptance. Moreover, to further improve the quality of the paper, I also recommend the authors carefully consider the issues I raised.

---

> ### Author Rebuttal · Authors · 2025-07-30
>
> Dear Reviewer RULe,
>
> Thank you for the positive comments and we appreciate the advices.
>
> **Regarding Weakness 1:**
> It is true that an adversary could fine-tune the model to improve its performance on specific tasks.
> Our assumption here was intended to convey that any modification to the model by the adversary could potentially cause errors in fingerprint verification: the higher the error rate, the lower the robustness.
> Therefore, a robust fingerprint should be resilient even against the strongest possible attacks.
> We acknowledge that our original statement was not clear enough and we would make appropriate clarification on this point in the final version.
>
> **Regarding Weakness 2:**
> We made this assumption and derivation to maintain generality, i.e., we focused on the primary and most difficult setting in which each model is equally likely to be stolen.
> In cases where the suspicious model is likely to be derived from a certain subset of all candidate models, the discussions in our paper maintain valid after replacing $F$ with this subset.
> When the suspicious model is more likely to be some specific model within $F$, the entropy of the copyright tracing event is strictly smaller than $\log_{2}|F|$, so our conclusions on the minimal number of triggers remain a pessimistic yet valid bound.
> In future revisions, we will supplement our discussion to include the influence of prior biases.
>
> **For the last concern**, line 130-131 mean that both the output on the $n$-the trigger ($\phi_{n}$) and the outputs on all triggers ($\Phi$, which is composed of $\phi_{1},\phi_{2},\cdots,\phi_{N}$) are random variables, which may be grammatically misleading but is not a typo.
>
>
> Thanks again for your insightful suggestions.

---

> > ### Comment · Reviewer_RULe · 2025-08-08
> >
> > Thank you for your rebuttal. It addresses my concerns.  I will keep my score.

---

### Official Review · Reviewer_9qe1 · 2025-07-03

**Clarity:** 3
**Significance:** 3
**Originality:** 2
**Rating:** 3
**Confidence:** 4

**Summary:**

This paper explores an information-theoretic approach to enhance the uniqueness of deep neural network (DNN) fingerprints for copyright tracing. The authors propose selecting fingerprinting triggers by maximizing conditional mutual information, aiming to improve the uniqueness rate of fingerprints. Experiments on datasets including MNIST, FashionMNIST, CIFAR-10, and ImageNet validate the proposed approach.

**Questions:**

1. How does the fingerprinting method affect model accuracy, and can you provide a quantifiable cost-benefit analysis?

2. Why were comprehensive comparisons with state-of-the-art fingerprinting methods (such as MetaFinger or Characteristic Examples) not included experimentally?

3. How do you anticipate your method scaling to more complex, large-scale real-world datasets and models?

**Ethical Concerns:**

["NO or VERY MINOR ethics concerns only"]

**Final Justification:**

While the paper is clearly written and presents interesting ideas, its applicability is still limited due to the lack of validation on more complex, realistic scenarios—especially regarding model performance under adversarial conditions, model modifications, and fine-tuning. I appreciate the authors’ hard work and contributions, but these gaps mean the proposed method is not yet ready for real-world deployment, so I recommend keeping the current rating.

**Limitations:**

No. The authors have not adequately addressed the potential negative societal impacts or thoroughly discussed limitations beyond adversarial robustness.

Suggestions for improvement:
1. Explicitly discuss any potential negative impacts, such as misuse or unintended consequences related to model fingerprinting and intellectual property tracing.

2. Provide a deeper exploration of the broader ethical implications of deploying this technology in practical settings.

3. Clearly outline the limitations associated with scalability, robustness in diverse real-world conditions, and potential accuracy degradation due to fingerprinting.

**Quality:**

3

**Strengths And Weaknesses:**

Strengths
+ The theoretical analysis leveraging information theory is solid, and the greedy trigger-selection algorithm is clear and well-justified.
+ The paper is generally well-structured, with clear explanations and systematic presentation of concepts and experiments.
+ Using information theory for optimizing fingerprints has some novelty, although similar ideas about optimizing robustness have been explored before.

Weaknesses
- Limited Realism in Experimental Setting: The experiments primarily utilize simpler, smaller-scale datasets like MNIST, FashionMNIST, and CIFAR-10. Although ImageNet is included, experiments on more complex and realistic scenarios remain limited.

- Insufficient Comparison with State-of-the-Art: While several existing fingerprinting methods are listed, the paper lacks comprehensive comparative experiments against these methods, notably missing direct experimental comparisons to established state-of-the-art methods such as MetaFinger or Intrinsic Examples.

- Lack of Accuracy-Cost Analysis: The paper does not explicitly analyze or quantify the cost associated with accuracy degradation due to fingerprinting, an essential consideration for practical deployment.

- Incomplete Coverage of Relevant Literature: Important related works such as "Characteristic Examples: High-robustness, Low-transferability Fingerprinting of Neural Networks" (Wang et al., IJCAI 2021) and "An Overview of Trustworthy AI" (Zheng et al., IEEE 2024) were not sufficiently considered or compared against in the paper. Specifically, the concept of "Low-transferability" discussed in Wang et al. (2021) closely aligns with the notion of "uniqueness" explored in this paper, yet the comparison is absent.

---

> ### Author Rebuttal · Authors · 2025-07-30
>
> Dear Reviewer 9qe1,
>
> Thank you for the insightful comments.
>
> We summarize your concerns into five aspects: **Settings of datasets & models**, **Settings of comparision baselines**, **Accuracy-cost analysis**, **Relevant literature**, **Limitations & ethical considerations**.
>
> **Settings of datasets & models (Weakness 1, Question 3):**
> In addition to image classification, we also conducted experiments on generative language models.
> This is mentioned in *Remark 3, Sec.3.3*, results are reported in *Appendix B*.
>
> **Settings of comparison baselines (Weakness 2, Question 2):**
> Our method is not a stand-alone fingerprinting scheme.
> Instead, it is a universally application optimization scheme that improves the uniqueness of DNN fingerprinting schemes (either established schemes or future schemes), and is not a direct competitors to any DNN fingerprinting scheme.
> To the best of our knowledge, we are the first to study the uniqueness of trigger set-based fingerprints for black-box scenarios in large-scale distribution settings as an individual problem.
> MetaFinger has been cited as Reference [15], and we demonstrated the performance of our scheme with it under the name *Adv-1*.
> Intrinsic Examples focuses on optimizing the fingerprint embedding stage, and is out of the scope of our study, which focus on selection of triggers without tuning the original model.
>
> **Accuracy-cost analysis (Weakness 3, Question 1):**
> Our method modifies the trigger selection process and does not modify the underlying model.
> Therefore, it introduces zero performance degradation.
> The trigger-set-based fingerprinting methods that we optimize do not modify the model either, so there is no accuracy loss involved.
> The overhead caused by trigger selection has been detailed in *Sec.4.4, Fig.7*.
>
> **Relevant literature (Weakness 4):**
> The low transferability of Characteristic Examples pertains to the fingerprint of a single model, while our work focuses on the uniqueness of fingerprints in distinguishing between models in large-scale deployments, we would properly cite this work in the prelimary section.
> "An Overview of Trustworthy AI" is a survey, yet it neither specificifies the uniqueness of fingerprint nor provide solution, so we did not include it in our related works.
>
>
> **Limitations & ethical considerations (Limitations):**
> The limitations of this work is that we have not applied it to multi-model tasks, and we mention this by the end of *Sec.5*.
> The work described in this paper aims to advance the field of artificial intelligence in a positive way. While our research may have broader societal implications, these considerations have not been the primary focus of the current discussion. We will include a dedicated examination of these aspects in future revisions.
>
> Thanks again for your valuable feedback and careful review.
>
> If you have any other concerns, we would be happy to discuss them.
>
> If not, we would be grateful if you could consider increasing your evaluation of our work.

---

> > ### Comment · Reviewer_9qe1 · 2025-08-06
> >
> > I believe the paper still lacks a thorough comparison with related works, especially other model fingerprinting methods that has no impact on the reported model accuracy. Additionally, there is insufficient robustness analysis. For example, in related work such as Characteristic Examples, robustness was evaluated under model modifications—this aspect is missing from the current paper.

---

> > > ### Author Response · Authors · 2025-08-06
> > >
> > > Thank you for the comments.
> > >
> > > **Regarding your first comment:**
> > >
> > > We emphasize that the method proposed in this paper is not a DNN fingerprinting scheme.
> > > Instead, it is a module that can be plugged into any DNN fingerprinting schme to improve the uniqueness.
> > >
> > > To this end, it is impossible to compare our module with other DNN fingerprinting schemes.
> > > Alternatively, what can be compared is the performance of DNN fingerprinting schemes before (in solid lines) and after (in dashed lines) adopting our scheme in Fig. 4.
> > >
> > > Following this reasoning, we considered several representative schemes for empirical studies and reported the results in the maunscript:
> > >
> > > [1] *Protecting intellectual property of deep neural networks with watermarking, in Proceedings of the 2018 on Asia conference on computer and communications security, 2018.*
> > >
> > > [2] *Robust watermarking of neural network with exponential weighting, in Proceedings of the 2019 ACM ASIACCS, 2019.*
> > >
> > > [3] *Rai 2: Responsible identity audit governing the artificial intelligence, in 2022 Network and Distributed System Security Symposium, 2022.*
> > >
> > > [4] *Certified neural network watermarks with randomized smoothing, in International Conference on Machine Learning. PMLR, 2022.*
> > >
> > > [5] *Afa: Adversarial fingerprinting authentication for deep neural networks, Computer Communications, vol. 150, pp. 488–497, 2020.*
> > >
> > > [6] *Ipguard: Protecting intellectual property of deep neural networks via fingerprinting the classification boundary, in Proceedings of the 2021 ACM ASIACCS, 2021.*
> > >
> > > [7] *Fingerprinting classifiers with benign inputs, IEEE Transactions on Information Forensics and Security, vol. 18, pp. 5459–5472, 2023.*
> > >
> > > Specifically, we used their patterns of triggers and do not fine-tune the models, so the performance of model remains uniformly intact.
> > >
> > > **Regarding your second comment:**
> > >
> > > You mentioned a counterexample that used model modification to evaluate the scheme's robustness and claimed that our paper lack such an aspect.
> > >
> > > The entire *Sec.4.2.2 Adversarial Setting: $\epsilon>0$* is devoted to discussions on robustness.
> > >
> > > Concretely, we eveluated the performance of our method under fine-pruning (this point is highlighted in Sec.4.1, line 233).
> > > And fine-pruning is well recognized as a representative **model modification**, please refer to Table III in [8] (the third row).
> > >
> > > It is true that more evaluations and discussions regarding robustness can be conducted and reported, but the discussions and analyses about robustness is not missing in this manuscript.
> > >
> > > [8] *SoK: How Robust is Image Classification Deep Neural Network Watermarking? IEEE S&P, 2022.*

---

### Comment · Area_Chair_Ycpw · 2025-08-06
**Please read the author's rebuttal and discuss with the author**

Dear Reviewers,

The paper has received mixed reviews. As the deadline for the discussion period approaches (August 8), please take a moment to read the authors' responses and reply to them at your earliest convenience if you haven't already done so. It’s important to give the authors ample time to respond, so please avoid waiting until the last minute.

If you plan to maintain your current rating of the paper, kindly respond to the authors to confirm your support or any unresolved concerns. This feedback will be very helpful to the authors.

Thank you,
AC

---

### Note · Authors · 2025-08-11

We thank all reviewers and AC for their efforts devoted to discussing and improving our work.

It was observed that some concerns from Reviewer 9eq1 remained unsolved, especially regarding the comparisons between other baselines and the consideration about robustness.

To prevent misunderstandings within these concerns, we gave an explanation that highlighted the unique contributions and the reasons behind the experimental settings of this paper (see our official comments at 06 Aug 2025, 15:29).

**On one hand**, our scheme is is not a DNN fingerprinting scheme.
Instead, it is a module that can be plugged into any DNN fingerprinting scheme to improve the uniqueness, and it is impossible to compare such a module with other end-to-end DNN fingerprinting schemes.
We can only compare the performance of DNN fingerprinting schemes (we conducted experiments on several updated schemes) before and after adopting our module, the results are shown in solid lines and dashed lines respectively in Fig. 4.

**On the other hand**, we eveluated the performance of our method under fine-pruning (this is mentioned in Sec.4.1, line 233), which is well recognized as a representative model modification method (please refer to Table III, the third row in *SoK: How Robust is Image Classification Deep Neural Network Watermarking? IEEE S&P, 2022.*).
Therefore, the discussions and analyses about robustness is not absent in this manuscript.

We hope that this explanation could address the remained concerns and benifit any further internal discussions.

Thank you again for all the efforts and insightful comments.

---

### Decision · Program_Chairs · 2025-09-17

**Decision:**

Accept (poster)

**Comment:**

The paper aims to enhance figureprint trigger selection for DNNs by choosing samples with the highest marginal mutual information. The approach is information-theory inspired and can determine an optimal number of triggers to reliably identify models. The contribution from an information theory perspective to the model fingerprinting process is novel, and the experiments are also comprehensive on multiple models and datasets.

The reviewers and the AC are generally positive about the paper, and some concerns are addressed during the discussion period. The main issue remaining is the lack of discussion of some related work. When the AC recommends acceptance of this paper, the authors should discuss related work in a more comprehensive way and explain how this work can be specifically integrated into these existing schemes. This will reduce the confusion from readers.